# Putative Biosynthesis of Talarodioxadione & Talarooxime from *Talaromyces stipitatus*

**DOI:** 10.3390/molecules27144473

**Published:** 2022-07-13

**Authors:** Ahmed J. al Fahad

**Affiliations:** National Center for Biotechnology, Life Science & Environment Research Institute, King Abdulaziz City for Science and Technology, Riyadh 11442, Saudi Arabia; ajlfahad@kacst.edu.sa; Tel.: +966-114-883-555

**Keywords:** polyketides, polyesters, secondary metabolites, biosynthetic gene cluster, tryptophan metabolism, structure elucidation

## Abstract

Polyesters containing 2,4-dihydroxy-6-(2-hydroxypropyl)benzoate and 3-hydroxybutyrate moieties have been isolated from many fungal species. *Talaromyces stipitatus* was previously reported to produce a similar polyester, talapolyester G. The complete genome sequence and the development of bioinformatics tools have enabled the discovery of the biosynthetic potential of this microorganism. Here, a putative biosynthetic gene cluster (BGC) of the polyesters encoding a highly reducing polyketide synthase (HR-PKS) and nonreducing polyketide synthase (NR-PKS), a cytochrome P450 and a regulator, was identified. Although talapolyester G does not require an oxidative step for its biosynthesis, further investigation into the secondary metabolite production of *T. stipitatus* resulted in isolating two new metabolites called talarodioxadione and talarooxime, in addition to three known compounds, namely 6-hydroxymellein, 15G256α and transtorine that have never been reported from this organism. Interestingly, the biosynthesis of the cyclic polyester 15G256α requires hydroxylation of an inactive methyl group and thus could be a product of the identified gene cluster. The two compounds, talarooxime and transtorine, are probably the catabolic metabolites of tryptophan through the kynurenine pathway. Tryptophan metabolism exists in almost all organisms and has been of interest to many researchers. The biosynthesis of the new oxime is proposed to involve two subsequent N-hydroxylation of 2-aminoacetophenone.

## 1. Introduction

Filamentous fungi are important sources of structurally diverse and biologically active secondary metabolites, including polyketides, non-ribosomal peptides, and terpenes [1]. The fungal genus *Talaromyces* produces a broad spectrum of bioactive natural products of interest to many scientists [2,3]. Polyesters containing 2,4-dihydroxy-6-(2-hydroxypropyl)benzoate and 3-hydroxybutyrate moieties are polyketide-type compounds isolated from many filamentous fungi such as *Penicillium verruculosum* [4,5], *Acremonium butyri* [6], *Talaromyces flavus* [7,8], *Menisporopsis theobromae* [9], *Ascotricha* sp. [10] and *Hansfordia sinuosae* [11]. These polyketides are produced as linear or macrocyclic structures with different patterns of subunit assembly and potential biological activities, such as antimalarial, antifungal, antibacterial and anticancer [12,13]. In particular, macrocyclic polyesters are more bioactive than their opened-structure counterparts because cyclic compounds retain rigidity and do not lose entropy upon interacting with their targeted enzymes [14,15]. 

The biosynthesis of the fungal polyesters requires two enzymes, an HR-PKS and NR-PKS [16]. Early study of the biosynthesis of polyesters showed the incorporation of [1-^13^C]- and [2-^13^C]-labelled acetate into the structure of a macrocyclic polylactone menisporopsin A, which confirmed the polyketide origin of such a fungal polyester [17]. Later, a gene cluster found in *Menisporopsis theobromae* BCC 4162 containing two polyketide synthase genes, namely *men1* an HR-PKS and *men2* an NR-PKS, was identified. Heterologous coexpression of *men1* and *men2* in a fungal host *Aspergillus oryzae* led to the production of a macrocyclic polyester, ascotrichalactone A, and other byproducts such as 6-hydroxymellein **6** and orthosporin [16]. The characterization of Men1 and Men2 proved that the biosynthesis of the fungal polyester did not require additional enzymes and highlighted the unusual mechanisms of interdomain interaction of Men2 during the process of subunit assembly.

*Talaromyces stipitatus* is known to yield polyketides, predominantly tropolones [18,19]. Recently, a polyester named talapolyester G 2 containing 2,4-dihydroxy-6 (2-hydroxypropyl)benzoate and 3-hydroxybutyrate moieties was isolated from this organism [20]. Fortunately, the complete genome sequence of *T. stipitatus* and the development of bioinformatics tools have enabled the discovery of its biosynthetic potential [21]. Here, a new cyclic polyester named talarodioxadione 1 and other known polyketides, namely 6-hydroxymellein 6 [22] and the polyester 15G256α 3 [23], which have never been reported from *T. stipitatus*, were isolated and characterized. In addition, the bioinformatic analysis revealed a potential gene cluster for *T. stipitatus* polyesters biosynthesis that consists of four genes, including an HR-PKS, NR-PKS, a cytochrome P450, and a regulatory gene. The secondary metabolic investigation supported the putative gene cluster of polyester and revealed the production of some catabolic metabolites of tryptophan, such as a new oxime called talarooxime 5 and the quinolone transtorine 4 [24] (Figure 1).

## 2. Materials and Methods

### 2.1. Fermentation and Extraction

*Talaromyces stipitatus*, (Thom) C.R. Benjamin, CBS 349.72 was provided by the Leibniz Institute DSMZ (Germany). Two liters of the fungal culture were grown in Czapek-Dox medium supplemented with tryptone (20 g/L sucrose, 10 g/L tryptone, 2 g/L NaNO_3_, 2 g/L KCl, 1 g/L K_2_HPO_4_, 0.5 g MgSO_4_·7H_2_O, 0.01 g FeSO_4_·7H_2_O) for 7 days at 28 °C and shaken at 200 rpm. The culture broth was acidified to pH 5 using diluted HCl solution and then vacuum filtered using normal filter paper. The secondary metabolites were extracted using a separation funnel and ethyl acetate, then dried over anhydrous magnesium sulfate. The drying agent was filtered, and the organic solvent was evaporated using a rotary evaporator at 40 °C. Around 1.8 g of crude extract was obtained.

### 2.2. Isolation of Metabolites

The ethyl acetate crude extract was subjected to flash chromatography using silica gel high-purity grade, pore size 60 Å, 220–440 mesh particle size, 35–75 μm particle size, and 50% ethyl acetate and dichloromethane as a mobile phase. Colorless fractions were firstly eluted and combined together. Upon evaporation of the organic solvent, pale-yellow residue, which contained **1**, **2**, **4**, **5** and **6** metabolites, was afforded, whereas **3** was directly eluted with around 450 mL 100% of ethyl acetate. After dryness, the fraction residues were redissolved in methanol for further purification using preparative HPLC (Agilent 1260 Infinity II Analytical-Scale LC Purification System), which comprises of 1260 Infinity II preparative binary pump (G7161A), 1260 Infinity II preparative autosampler(G7157A), Infinity II diode array detector (G7115A), 1260 Infinity II Analytical Fraction Collector (G1364E). The purification was performed with an Agilent ZORAX SB-C18 column (9.4 mm × 250 mm, 5 μm) at room temperature, flow rate: 3 mL/min, a mobile phase composed of Milli- deionized water +0.1% formic acid and HPLC grade acetonitrile +0.1% formic acid and runs of 30 min (3 min, 5% B; 20 min, 5–98% B; 4 min, 98% B; 1 min 98–5%, 3 min 5% B). HPLC-fractions of every compound were combined and dried under a nitrogen stream to afford 8.0 mg of talaroxime **5** (*t_R_*: 10 min), 3.9 mg of 6-hydroxymellein **6** (11.7 min), 5.4 mg of transtorine **4** (12.3 min), 9.5 mg of talarodioxadione **1** (14.2 min), 6.3 mg of talapolyester G **2** (15.5 min) and 8.2 mg of 15G256α **3**, (16.8 min). 

### 2.3. Structure Elucidation

Purified compounds were characterized from the analyses of 1D and 2D NMR spectra which were obtained on a 600 MHz Bruker Avance III spectrometer. The high-resolution mass spectrometry and the isotopic distribution (ID) of compounds were obtained using a Q-Exactive Orbitrap mass spectrometer (Thermo Scientific, San Jose, CA, USA). Fourier transform infrared (FT-IR) spectra were recorded on Nicolet iS20 FTIR Spectrometer (Thermo Scientific, Waltham, MA, USA). Optical rotations were determined at 589 nm using a Bellingham and Stanley ADP400 polarimeter.

### 2.4. Spectroscopic and Spectrometric Data

Talarodioxadione **1**: white crystals; ${\left[\alpha \right]}_{D}^{22}$ −33.1 (*c* 0.1, MeOH), IR (neat): *v*_max_ 3196, 3082, 1627, 1476, 1253, 1167 cm^−^^1^; ^1^H-NMR (CDCl_3_, 600 MHz) *δ* (ppm) = 6.37 (s, 1H), 6.19 (d, *J* = 2.2 Hz, 1H), 5.30 (dt, *J* = 6.9, 3.5 Hz, 1H), 5.20 (ddd, *J* = 10.8, 6.8, 3.8 Hz, 1H), 3.66 (dd, *J* = 14.3, 3.8 Hz, 1H), 2.86 (t, *J* = 11.3 Hz, 1H), 2.51–2.47 (m, 2H), 1.68 (d, *J* = 6.6 Hz, 3H), 1.03 (d, *J* = 6.7 Hz, 3H); ^13^C NMR (150 MHz, CDCl_3_) *δ* (ppm) = 171.2, 170.7, 160.2, 159.6, 139.3, 113.0, 110.6, 102.5, 74.6, 70.9, 41.3, 39.3, 21.00, 18.2; HR-ESI-MS at *m*/*z* 281.1017 [M + H]^+^, (calcd. for C_14_H_17_O_6_: 281.1025).

Talarooxime **5**: colorless crystals, IR (neat): *v*_max_ 3402, 2970, 2880, 2742, 2359, 1652, 1525, 1451, 1451, 1281, 765 cm^−1^; ^1^H-NMR (DMSO-*d_6_*, 600 MHz) *δ* (ppm) = 9.72 (s, 1H), 9.29 (s, 1H), 7.66 (dd, *J* = 7.9, 1.6 Hz, 1H), 6.93 (td, *J* = 8.1, 1.6 Hz, 1H), 6.84 (dd, *J* = 8.1, 1.5 Hz, 1H), 6.75 (td, *J* = 7.9, 1.5 Hz, 1H), 2.09 (s, 3H); ^13^C NMR (DMSO-*d6*, 150 MHz) *δ* (ppm) = 168.9, 147.8, 126.4, 124.6, 122.3, 118.9, 115.9, 23.6. HR-ESI-MS at *m/z* 152.07061 [M + H]^+^ (calcd. for C_8_H_10_O_2_: 152.07115).

The NMR data of the metabolites **2**, **3**, **4** and **6** agreed with the literature. Furthermore, 1D and 2D-NMR spectra of all metabolites are presented in the Appendix A (Appendix A).

### 2.5. Bioinformatics Analysis

The amino acid sequences of Men1 (CAB3277415.1) and Men2 (CAB3277416.1) were used as queries to perform BLASTP sequence similarity searching against the *T. stipitatus* non-redundant protein sequence database. The algorithm parameters were as follows: the substitution matrix used was BLOSUM62, gap opening penalty 11, gap extension penalty 1, expectation value (E-value) 10^−10,^ and compositional matrix adjust method [25]. Conserved domain analyses were performed using the simple modular architecture research tool (SMART) [26] and SnapGene software.

## 3. Results and Discussion

### 3.1. Structure Elucidation

Talarodioxadione **1** was isolated as white crystals. Its molecular formula C_14_H_16_O_6_, which indicates 7 degrees of unsaturation, was established from HR-ESI-MS (positive mode) at *m*/*z* 281.1017 [M + H]^+^, (calcd. for C_14_H_17_O_6_: 281.1025). Additional confirmation was obtained from the isotopic distribution (ID), which matched the simulated ID of the proposed molecular formula. The ^1^H and ^13^C-NMR chemical shifts of compound **1** were similar to that reported for the fungal polyesters (Table 1). 

The ^1^H NMR in CDCl_3_ of **1** showed only two aromatic protons in *meta* position to each at δ_H_ (ppm) 6.37 and 6.19, which indicated the presence of a tetra-substituted benzene ring, whereas the 3-hydroxybutyrate moieties were recognized from the presence of two oxygenated methine protons at δ_H_ (ppm) 5.30 and 5.20, two methylenes at δ_H_ (ppm) 3.66, 2.86, and 2.49 (2H overlapped), and two methyl groups resonated as duplets at δ_H_ 1.68 and 1.03. One dimensional Nuclear Overhauser Effect Spectroscopy (1D-NOESY) spectra were indicative of the correlation between the aromatic proton H-5 and the aliphatic protons H_a,b_-7. Other ^1^H-^1^H space couplings were detected between H-7/H8, H-7/H9, H-8/H-9, H-11/H12, H-12/H13, and H-11/H-13. The ^13^C-NMR spectrum showed 14 carbons assigned based on Heteronuclear Single Quantum Coherence (HSQC) and Heteronuclear Multiple Bond Connectivity (HMBC) analyses. The HMBC spectral data suggested the cyclic structure of **1**. A Key HMBC ^1^H-^13^C long rang correlation was detected between H-12/C-14. More correlations supported the predicted structure of **1** were observed from H-3 to C1/ C-2/C-5, from H-5 to C-1/C-3/C-4/C-7, from H_ab_-7 to C-5/C-6/C-8, from H-9 to C-7/C-8, from H_ab_-11 to C-10/C-12/C-13, from H-12 to C-14, and from H-13 to C-10/C-11/C-12 (Figure 2).

Talarooxime **5** was obtained as colorless crystals. The molecular formula of **5** was determined from the analysis of HR-ESI-MS and ID. The HR-ESI-MS positive mode showed an ion at *m*/*z* 152.07061 [M + H]^+^ (calcd. for C_8_H_10_O_2_: 152.07115). The NMR experiments were performed in DMSO-*d_6_* to enable the detection of exchangeable protons. Two broad singles of the two hydroxyl groups appeared at δ_H_ (ppm) 9.72 and 9.29, whereas the four aromatic protons resonated at δ_H_ (ppm) 7.66 (dd), 6.93 (td), 6.84 (dd) and 6.75 (td). 1D-NOESY experiment helped assign the configuration of **5**. Clear NOESY correlations between the hydroxyl groups and the aromatic protons were observed, suggesting *trans*-conformation around the external double bonds (Figure 2). ^13^C-NMR showed eight signals which were assigned from HSQC and HMBC experiments. Protonated carbons showed cross-couplings in the HSQC spectrum at δ_c_ (ppm): H-3/C-3 (122.3), H-4/C-4 (118.9), H-5/C-5 (124.6), H-6/C-6 (115.9) and H-8/C-8 (23.6). The designation of C-1, C-2 and C-7 were deduced from the HMBC spectrum at δ_c_ (ppm) 147.8, 126.4 and 168.9, respectively, as the ^1^H-^13^C correlations were detected from N-OH to C-1/C-2/C-6, from -CH_3_ to C-7, and from C-OH to C-2/C-3. 

### 3.2. Identification of Talaromyces Stipitatus Polyesters Biosynthetic Gene Cluster

The polyesters isolated from *T. stipitatus* consist of 2,4-dihydroxy-6-(2-hydroxypropyl)benzoate and 3-hydroxybutyrate subunits, which were predicted to derive from an HR-PKS and NR-PKS homologous to Men1 and Men2. Indeed, protein Blast searches using the amino acid sequence of Men1 against the *T. stipitatus* protein database revealed a highly homologous HR-PKS (XP_002488696) at the top hit with 59% identity and 73% similarity, which is over twice higher similarity than the second top hit. The Men2 query showed an NR-PKS (XP_002488697) that shares a significant similarity, with 67% identity and 80% similarity compared to the second top hit (Appendix A). Interestingly, the two corresponding genes of XP_002488696 and XP_002488697, which were named *tpeA* and *tpeB*, existed together in a gene cluster with a size of around 27.5 kb. The identified gene cluster also contains a cytochrome P450 oxygenase (*tpeC*) and a regulator protein (*tpeD*) (Figure 3).

The conserved domain analysis of TpeA and TpeB was performed using the SMART protein domain analysis tool. Both enzymes showed similar domain architecture to Men1 and Men2, respectively (Appendix A). TpeA consists of the following domains: i.e., ketosynthase (KS), malonyl-CoA:ACP transacylase (MAT), dehydratase (DH), enoyl reductase (ER), ketoreductase (KR), and an acyl carrier protein (ACP). Similar to what had been observed for Men1, TpeA has the structure of an HR-PKS but could only catalyze the reduction in the β-carbonyl intermediate since the DH and ER domains are non-functional. Active DH domains in highly reducing (HR)-PKSs and partially reducing (PR)-PKSs have the His/Asp catalytic diad in the conserved motifs HXXXGXXXXP and DXXX(Q/H) to adopt the double-hotdog fold [27]., but TpeA has the motifs H^980^XXXSXXXXP^989^ and E^1169^XXXQ^1173^, indicating the replacement of Asp with Glu which may not be enough as indicative for non-functional DH^0^ domain [28,29]. However, the conserved LPFXW motif and Arg residue, which were considered essential for the interaction with the ACP domain, are also not present in TpeA (Appendix A) [16,30]. In addition, the ER domain lacks the NADPH binding motif (GGVG) and has instead the G^1945^AVG^1948^ motif, suggesting the inactive ER^0^ domain [31]. 

TpeB has a typical fungal NR-PKS domains, including a starter-unit acyltransferase (SAT), ketosynthase KS, MAT, product template (PT), tandem ACP doublet and thioesterase (TE) (Table 2) [32,33,34]. The similarity of TpeA and TpeB domains to their counterparts in Men1 and Men2 are presented in (Appendix A).

### 3.3. The Biosynthesis of Talaromyces Stipitatus Polyesters

The biosynthesis of the polyesters probably starts with the formation of the diketide 3-hydroxybutyryl-S-ACP catalyzed by the HR-PKS (TpeA). The acceptance of 3-hydroxybutyryl by the NR-PKS (TpeB) would initiate further elongation and cyclization, catalyzed by KS and PT, respectively, to form 2,4-dihydroxy-6-(2-hydroxyn-propyl)benzoyl-S-ACP intermediate. The TE domain could catalyze lactonization at this step to yield 6-hydroxymellein **6** as a derailment product. This polyketide was also produced by transformants of *A. oryzae* expressing *men1* and *men2*. Similarly to the biosynthetic steps of ascotrichalactone A, the polyesterification process maybe occurs when additional molecules of 3-hydroxybutyryl are transferred to the NR-PKS [16]. Following the first esterification step, an intramolecular cyclization catalyzed by the TE domain would give talarodioxadione **1**, whereas the ethyl esterification of talapolyester G **2** perhaps happens spontaneously (Figure 1a). Accumulated evidence showed that TE domains of iterative polyketide display selectivity for intermediates offloading with constant monitoring for the shape and size of intermediates during the elongation process until completion [35]. However, smaller molecules could escape further elongation during the assembly process, depending on the selectivity of the TE domain. Elegant experiments conducted by Xu et al. investigated the effect of TE domain swap on the programming of NR-PKSs and the production of unnatural products. The outcome of such experiments showed the importance of the TE domain in controlling product release and its capability for macrocyclization, hydrolysis and transesterification [36].

The mechanism of esterification is ambiguous. It was hypothesized that esterification and cyclolactonization are catalyzed by the TE domain as the reduced diketide intermediate 3-hydroxybutyryl is passed to the TE for the first esterification reaction with the 2,4-dihydroxy-6-(2-hydroxyn-propyl)benzoyl-S-ACP subunit and further esterification reactions happen through interactions between TE and ACP_2_ domains. Although the low-resolution small-angle X-ray scattering (SAXS) model of Men2 ACP_1_-ACP_2_-TE showed a great flexibility beads-on-a-string configuration, there was no interaction between ACP and TE domain was detected [16,37]. Another investigation using a high-resolution NMR structural and biophysical analysis of a closely resembled NR-PKS tandem ACP domains (PigH ACP_1_-ACP_2_) from prodigiosin biosynthesis revealed two distinct conformers, bent and extended. In addition, an interaction between the ACP domains and the ACP cross-liker has also been observed, which may suggest the importance of the linker sequence in the biosynthetic programming of NR-PKS [38]. Conceivably, the double ACP domains may require specific interactions with the linker sequences to facilitate the interaction with TE domain and esterification reaction. However, the mechanisms of NR-PKS possessing tandem ACP domains are still not fully understood and are difficult to predict [39].

Since talapolyester G **2** does not require an oxidative step, the existence of other polyesters was predicted. The investigation led to identifying the macrocyclic polyester 15G256α, which was also produced by other filamentous fungi, *Hypoxylon oceanicum* LL-15G256 [23], and *Talaromyces flavus* [40]. The biosynthesis of 15G256α was previously proposed to involve the hydroxylation of an inactive methyl group since potential intermediates such as 15G256β and 15G256β-2 were isolated for the same fungus [41]. This biosynthetic proposal for 15G256α is in agreement with the discovered BGC of *T. stipitatus* polyesters as the P450 (TpeC) is the best candidate for catalyzing this oxidative step (Figure 1b). Although 15G256β and 15G256β-2 have not been detected in the secondary metabolites production of *T. stipitatus*, presumably due to the fast oxidation reaction, further subunit assembly catalyzed by TpeB could offer the two substrates. 

### 3.4. The Biosynthesis of Talarooxime

Talarooxime is a possible catabolic product of tryptophan. L-tryptophan is an essential amino acid for microorganisms, but a small fraction is utilized in protein biosynthesis, whereas an excess quantity is metabolized through the kynurenine pathway [42]. The kynurenine pathway is important for generating the cofactor nicotinamide adenine dinucleotide (NAD). However, there are many metabolites of tryptophan degradation, and one of them is 2-aminoacetophenone **10** [43], which is likely a substrate for talarooxime biosynthesis. The natural production of 2-aminoacetophenone came from converting kynurenine **8** to kynurenic acid **9** [44]. The latter is in an equilibrium with transtorine **4**, which was previously isolated from a plant organism, *Ephedra transitoria* [24]. Two subsequent N-hydroxylation of **10** could result in oxime formation and the yield of **5** (Figure 2). Similar oxidation has been reported in tyrosine and tryptophan metabolism [45]. For example, the cytochrome P450 (CYP79B2) was found to hydroxylate twice the amino group of tryptophan **7** to yield indole-3-acetaldoxime **11** [46]. Tryptophan metabolism via the kynurenine pathway exists in almost all living organisms. It has been linked to the progression of many human diseases such as cancer [47], HIV [48], and COVID-19 [49] as it prevents inflammation and contributes to the development of immunodeficiency. Therefore, the characterization of such a metabolite in the kynurenine pathway may provide an insight into medical research.

## 4. Conclusions

The fungal polyesters containing 2,4-dihydroxy-6-(2-hydroxypropyl)benzoate and 3-hydroxybutyrate moieties are interesting polyketides due to their potential biological activities and unusual biosynthesis. The shared production of some polyesters by many filamentous fungi suggests programmed esterification catalyzed by dedicated HR-PKS and NR-PKS. The fungus *T. stipitatus* continues to provide a biosynthetic understanding of fungal secondary metabolites. The investigation of *T. stipitatus* secondary metabolites resulted in the identification of new and unreported compounds, which led to the discovery of a putative BGC encoding all enzymes, namely TpeA (HR-PKS), TpeB (NR-PKS), TpeC (P450) and TpeD (regulator) required for the biosynthesis of 15G256α. The presented result may allow for domain swap experiments to greater understand the mechanism and programming that govern the iterative esterification process. In addition, tryptophan metabolism exists in almost all organisms and has been of interest to many researchers. Therefore, the characterization of talarooxime may help identify and understand such a metabolite’s effect on human health.

## Data Availability

All data are available upon request.

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
