# Peer review of "Putative Biosynthesis of Talarodioxadione & Talarooxime from Talaromyces stipitatus"

_molecules, 2022, doi:10.3390/molecules27144473_

Round 1

Reviewer 1 Report

- Minor:

1) Authors should explain names of the two new compounds: Talarodioxadinone and Talarooxime.

 2) Is it possible to determine the amount of these substances (mg per L of the nutrient medium)?

Author Response

1) Authors should explain names of the two new compounds: Talarodioxadinone and Talarooxime.

Probably your question proposes to revise the names. Thank you very much for this suggestion as I realized that I made a mistake in naming Talarodioxadinone as I intended to call it Talarodioxadione. Therefore, Talarodioxadinone was corrected to Talaroxadione in the manuscript and the supporting document.

The names indicate the source of the compound and give a hint about the chemical structures. For example, Talaro from Talaromyces, dioxa indicates the two oxygens in the ring, dione for the two carbonyl groups and oxime for the presence of the oxime functional group.

 2) Is it possible to determine the amount of these substances (mg per L of the nutrient medium)?

Yes, it is possible, but it is more common to use a volume of culture with the amount of fungal secondary metabolites, and I had used this in my previous publications.

https://doi.org/10.1073/pnas.1201469109

Reviewer 2 Report

Thank you for editing the article for re-consideration by Molecules. The structure elucidation of the isolated compounds is a good experimental evidence for the correct structures and their existence as intermediates or products in the proposed biosynthetic steps. However, almost all the isolated compounds are known, which may affect the novelty of the work. The authors used publicly available databases doing computational analysis to spot biosynthetic steps, which is still theoretical biology. Revising the work published at Molecules indicates that this article needs deeper experimental investigation. The article still lacks experimental evidence for the proposed pathway at molecular level, i.e at least experimental evidence for the expression pattern of the involved transcripts is needed.

Author Response

Comments and Suggestions for Authors

Thank you for editing the article for re-consideration by Molecules. The structure elucidation of the isolated compounds is a good experimental evidence for the correct structures and their existence as intermediates or products in the proposed biosynthetic steps. However, almost all the isolated compounds are known, which may affect the novelty of the work. The authors used publicly available databases doing computational analysis to spot biosynthetic steps, which is still theoretical biology. Revising the work published at Molecules indicates that this article needs deeper experimental investigation. The article still lacks experimental evidence for the proposed pathway at molecular level, i.e at least experimental evidence for the expression pattern of the involved transcripts is needed.

Thank you very much for your comments and suggestions. Besides the new compounds, the known compounds play an important role in understanding the biosynthesis of Talarodioxadione and Talarooxime.

National Center for Biotechnology Information (NCBI) and its tools is widely used in scientific research and many research article have been published using only bioinformatic evidence.

https://doi.org/10.3390/jof4030096

https://doi.org/10.3390/life11040356

https://doi.org/10.1371/journal.pone.0014657

https://doi.org/10.1002/cbic.202000025

https://doi.org/10.1186/1471-2164-15-983

https://doi.org/10.3389/fmicb.2020.01408

Reviewer 3 Report

The manuscript “Putative Biosynthesis of Talarodioxadinone & Talarooxime from Talaromyces stipitatus” from Ahmed J. al Fahad describes the investigation of the secondary metabolited produced by T. stipitatus secondary metabolites. This study resulted in the identification of new and unreported compounds, which led to the discovery of a putative BGC encoding all enzymes required for the biosynthesis of 15G256α.

The research herein presented may help identify and understand the effect of talarooxime on human health. On addition, the autor did a good job with the isolation and characterization of the compounds and the work could be of interest for researchers working on natural products isolation and structural elucidation.

On addition, the quality of the presentation is good and the Schemes and Figures are accurate and of good quality.

However, there are several points that need clarification before I could recommend the publication of the manuscript.

The information is not clearly presented and some relevant information is omitted in several procedures. Authors need to re-read the manuscript, and work to revise/re-write to improve clarity.

For example, in lines 58-66: The description of the work presented in the manuscript should be clear and concise and it should be written in the same order as the work is presented in the manuscript. Revise this paragraph taking into account the above considerations.

The experimental procedures should also be revised for clarity. It is crucial to present all the relevant information to assure the reproducibility of the method.

For example, the procedure for the chromatographic separation is not clearly presented. In line 84 author  states that 50% ethyl acetate and dichloromethane as a mobile phase. The, he writes that metabolite 3 is eluted with pure ethyl acetate. Is the mobile phase a gradient from ethyl acetate/dichloromethane 1:1 to ethyl acetate?

Line 95: where author say “HPLC-fractions of every compound were combined and dried under nitrogen blowing…” I assume fractions are evaporated and/or lyophilized first.

The, if the products are isolated just by evaporation/nitrogen drying, I seriously doubt a crystals are formed. What you probably have under those conditions is an amorphous solid. The lack of melting point data, reinforce my suspicions. The physical state of the products should be more clearly presented. If any of the products has been actually crystalized, melting points should be included.

Also, there are several other minor issues:

Line 65: transtorine is actually a quinolone

Line 97: 6-Hydroxymellein for 6-hydroxymellein

Line 119: 13C NMR missing

Table 1: what does a and b stand for? Should be explained in the table footnotes

In Table 1 and along the text: include the units for the chemical shifts (ppm)

Line 167: water added to confirm the exchangeable protons?

Author Response

Comments and Suggestions for Authors

The manuscript “Putative Biosynthesis of Talarodioxadinone & Talarooxime from Talaromyces stipitatus” from Ahmed J. al Fahad describes the investigation of the secondary metabolited produced by T. stipitatus secondary metabolites. This study resulted in the identification of new and unreported compounds, which led to the discovery of a putative BGC encoding all enzymes required for the biosynthesis of 15G256α.

The research herein presented may help identify and understand the effect of talarooxime on human health. On addition, the autor did a good job with the isolation and characterization of the compounds and the work could be of interest for researchers working on natural products isolation and structural elucidation.

On addition, the quality of the presentation is good and the Schemes and Figures are accurate and of good quality.

However, there are several points that need clarification before I could recommend the publication of the manuscript.

The information is not clearly presented and some relevant information is omitted in several procedures. Authors need to re-read the manuscript, and work to revise/re-write to improve clarity.

For example, in lines 58-66: The description of the work presented in the manuscript should be clear and concise and it should be written in the same order as the work is presented in the manuscript. Revise this paragraph taking into account the above considerations.

The paragraph was re-written accordingly.

Here, a new cyclic polyester named talarodioxadione 1 and other known polyketides, namely 6-hydroxymellein 6 [22] and the polyester 15G256α 3 [23], which have never been reported from T. stipitatus, were isolated and characterized. In addition, the bioinformatic analysis revealed a potential gene cluster for T. stipitatus polyesters biosynthesis that consists of four genes, including an HR-PKS, NR-PKS, a cytochrome P450, and a regulatory gene. The secondary metabolic investigation supported the putative gene cluster of polyester and revealed the production of some catabolic metabolites of tryptophan, such as a new oxime called talarooxime 5 and the quinoline alkaloid transtorine 4 [24] (Figure 1).

The experimental procedures should also be revised for clarity. It is crucial to present all the relevant information to assure the reproducibility of the method.

For example, the procedure for the chromatographic separation is not clearly presented. In line 84 author  states that 50% ethyl acetate and dichloromethane as a mobile phase. The, he writes that metabolite 3 is eluted with pure ethyl acetate. Is the mobile phase a gradient from ethyl acetate/dichloromethane 1:1 to ethyl acetate?

The phrase was corrected as follows:

whereas 3 was directly eluted with around 450 ml 100% of ethyl acetate.

I was not concerned about changing the mobile phase gradually as the collected fractions were re-submitted for further purification.

Line 95: where author say “HPLC-fractions of every compound were combined and dried under nitrogen blowing…” I assume fractions are evaporated and/or lyophilized first.

HPLC-fractions of every compound were combined and dried under a nitrogen stream.

The, if the products are isolated just by evaporation/nitrogen drying, I seriously doubt a crystals are formed. What you probably have under those conditions is an amorphous solid. The lack of melting point data, reinforce my suspicions. The physical state of the products should be more clearly presented. If any of the products has been actually crystalized, melting points should be included.

During slow evaporation, I observed the formation of crystals but soon they were stuck to the vials which were challenging to handle for a melting point measurement, especially with small amounts.

Also, there are several other minor issues:

Line 65: transtorine is actually a quinolone

It was corrected to the quinolone transtorine 4

Line 97: 6-Hydroxymellein for 6-hydroxymellein

It was corrected.

Line 119: 13C NMR missing

It was added.

Table 1: what does a and b stand for? Should be explained in the table footnotes

The sentence (The geminal protons are labelled a and b. ) was added in the table footnotes

In Table 1 and along the text: include the units for the chemical shifts (ppm)

The units for the chemical shifts (ppm) were added to the table and along with the text

Line 167: water added to confirm the exchangeable protons?

Water was not added to confirm the exchangeable protons.

Round 2

Reviewer 2 Report

The bioinformatic tool in the article under review starts by two sequences (i.e. Men1 CAB3277415.1 and Men2 CAB3277416.1) used as queries to BLASTP T. stipitatus NCBI protein database. The recent response by the author presents 6 articles claiming for possible publishing when the experimental section is based only on bioinformatic work. It was nice to go through these articles to find out that:

1) Except for the fifth article (BMC genomics), none of the articles is based on bioinformatic work only. Even for the one BMC genomics, the huge work and the journal focus match well for a good acceptance by the community.

2) None of the articles build their bioinformatic work on previously published sequence to simply surf the NCBI Blastp tool, as done in the current article.

The first two articles analyze whole genome (>7 Mbp) including new (in-house) annotation, which is a worth to publish work even if real expression study is not included.

The third article in the author response (https://doi.org/10.1371/journal.pone.0014657) used in addition real DNA extraction, amplification, cloning, and gene walking.

The fourth example (https://doi.org/10.1002/cbic.202000025) presents the bioinformatic work, DNA isolation and sequencing. Isolated compounds were quantified and tested for inhibition of the multidrug resistance transporter ABCG2, to interpret possible structure-activity relationships. Here, the article addresses different bio-dimension matching the journal topic (ChemBiochem journal).

The fifth article is from BMC genomic (https://doi.org/10.1186/1471-2164-15-983), it fits the journal focus by massive mining of 211 genomes

The last article is from Front.Microbiol (https://doi.org/10.3389/fmicb.2020.01408), where the journal focus let the authors publish their morphological and physiological characterization of the strains, phylogenetic reconstruction of the detected genes, antibiotic activities of the tested compounds. Obviously, this is broader work compared to the article under review.

According to the recent letter by the author, the last comment in my previous review is still open.

Author Response

Thank you for going through the articles, I am pleased that you positively saw the outcomes of these articles even though an expression study was not included.

Building my bioinformatic work on previously published sequences is actually a positive thing as the characterization of Men1 and Men2 supported my findings. The proposal of the biosynthesis of Talaromyces stipitatus polyesters was done in a proper methodology including bioinformatics, secondary metabolites profile investigation and literature reviewing all of which are in perfect agreement.

I believe that the manuscript is now in a good shape and will be of interest to many researchers, thanks to you and to the other reviewers for your valuable feedback which absolutely improved my manuscript and I am grateful for it.

Although TpeA and TpeB are significantly homologous to Men1 and Men2 respectively, I considered them putative. Therefore, I hope you accept the manuscript as a preliminary report without including the expression experiment.

This manuscript is a resubmission of an earlier submission. The following is a list of the peer review reports and author responses from that submission.

Round 1

Reviewer 1 Report

The manuscript described the chemical investigation of fungus Talaromyces stipitatus, which led to the isolation of two new compounds along with four known metabolites. Bioinformatic analysis of the genome sequence found the putative biosynthetic gene cluster, which is a homolog of men gene cluster (from menisporopsin A biosynthesis).

1) In the abstract or last paragraph of introduction should clearly mention the new and known compounds found in this study. It wasn't clear that if the only new compounds are 1 and 5.

2) In the last paragraph of introduction, all the known compounds should be cited, where compounds 2, 4 and 6 did not provide references.

3) NMR text form for compound 5 is understandable but compound 1 should provide a NMR table so that readers is able to grasp the meaning of NMR chemical shifts to structure at a glance.

4) The fungus was provided by Leibniz Institute DSMZ, the genome sequence of this fungus should be provided (accession number) as possible.

5) Scheme 1 (a) should be improved, this figure should be presented in comprehensive way, as shown in this ref "doi:10.1039/c8ob02773k".

6) In this chemical investigation, author did not observe the presence of orthosporin and ascotrichalctone A from purification or LCMS profile? Because expression of the homologs of Men1 and Men2 observed the production of these metabolites, as shown in this ref "doi:10.1039/c8ob02773k".

7) Tpe gene cluster has a P450 at adjacent downstream of TpeB, does the Men gene cluster also has a P450 in the flanking region?

8) S13 and S14 is better to include sequence alignment of Men1 and Men2 with TpeA and TpeB together with other related homologs. This will allowed readers to make meaningful insight about their conserve sequence and motifs.

9) This interactive PKS is definitely for the production of 3 but it is not strictly control, therefore smaller products are produced such as 1 and 6.

Reviewer 2 Report

Major compulsory revisions:      

1) This manuscript (Ms) contains misprints, mistakes in English grammar and in the writing style. I recommend that the authors should use some help of a native English speaker or send the Ms to an English Editing Service that proofreads scientific writing.

2) The Introduction is poorly written in this Ms, it looks more like a description of the obtained results. The authors need to describe in more detail the spectrum of known substances in the used fungus.

3) Authors should improve all legends for figures in Ms, e.g. in Fig. 1 and 5 include the names of the compounds, in Fig. 3, 4, and 5 explain all used abbreviations, etc.

4) Methodological features of bioinformatic analysis are poorly described. In my opinion, these data are too general, there is no experimental evidence of the data being discussed. Ms needs experimental data proving participation of tpeA, tpeB, tpeC, and tpeD genes in polyester biosynthesis (expression, overexpression).

5) As I understand authors described two new compounds: Talarodioxadinone and Talarooxime. Why are these substances so named? Exactly such substances have not been described before? Is it possible to determine the amount of these substances (mg per g of the fresh or dry fungus weight)?

- Minor:

6) Line 251-252: “pathway for talaroosime” correct to “pathway for talarooxime”.

7) In title correct “Putative Biosynthesis of Talarodioxadinone & Talarooxime from Talaromyces stipitatus” to “Biosynthesis of Talarodioxadinone and Talarooxime from Fungus Talaromyces stipitatus”.

Reviewer 3 Report

The article authored by Ahmed J. al Fahad proposed a putative gene cluster underlying the biosynthesis of Talaromyces stipitatus polyesters. The structure elucidation of one new compound and other known ones was the main tool to propose the biosynthetic sequence.

The materials and methods section is divided into 4 subsections about known fermentation process and the routine isolation of the compounds, followed by one short section for the preliminary blast steps of the databank proteins (from another group) or their domains. For the results section, except for the structure elucidation and their NMR supplementary spectra, all other results and their figures are proposed or hypothesized.

The structure elucidation of one new compound is not a sufficient feature for this article to be accepted. Unless experimental molecular evidence is provided for the proposed pathway, this article can be used as a short commentary for previously reported ones. I do not see that the provided data are sufficient for publication at molecules.